

# The biological function of m6A methyltransferase KIAA1429 and its role in human disease

Xiaoyu Zhang[1,*], Meng jiao Li[2,*], Lei Xia[3] and Hairong Zhang[4]

[1] Shandong University of Traditional Chinese Medicine, Jinan, China
[2] Liaocheng Vocational and Technical College, Liaocheng, China
[3] Shandong University of Traditional Chinese Medicine, Department of Pathology, Jinan, China
[4] Shandong Provincial Third Hospital, Department of Obstetrics and Gynecology, Jinan, China
[*] These authors contributed equally to this work.

## ABSTRACT

KIAA1429 is a major m6A methyltransferase, which plays important biological and pharmacological roles in both human cancer or non-cancer diseases. KIAA1429 produce a tumorigenic role in various cancers through regulating DAPK3, ID2, GATA3, SMC1A, CDK1, SIRT1 and other targets, promoting cell proliferation, migration, invasion, metastasis and tumor growth . At the same time, KIAA1429 is also effective in non-tumor diseases, such as reproductive system and cardiovascular system diseases. The potential regulatory mechanism of KIAA1429 dependent on m6A modification is related to mRNA, lncRNA, circRNA and miRNAs. In this review, we summarized the current evidence on KIAA1429 in various human cancers or non-cancer diseases and its potential as a prognostic target.

## INTRODUCTION

m6A methylation was first discovered in 1975, referring to RNA methylation at position N6 in adenosine (*Schmidt, Arnold & Kersten, 1975*). It is thought that m6A methylation is the most abundant and widespread epigenetic transcriptomic modification in eukaryotic mRNAs, accounting for 0.1% ∼0.4% of all adenosines (*Fu et al., 2014*). m6A modification usually occurs on the conserved sequence RRACH (where R represents G or A; H denotes A, C, or U) of 3′ untranslated region (3′UTR), long exonic region, and termination codon region (*Meyer et al., 2012*), where A is converted to m6A. In addition to mRNA, miRNA, lncRNA, circRNA, and snoRNA all have m6A modification sites, and their regulation involves almost all kinds of protein-coding and non-coding genes (*Ma et al., 2019*). In molecular mechanisms, m6A involves in almost all steps of RNA metabolism, including mRNAs translation (*Li et al., 2017*) , splicing (*Reichel, Köster & Staiger, 2019*), stabilization (*Wang et al., 2014*), output (*Fustin et al., 2013*) and folding (*Liu & Gregory, 2019*; *Liu et al., 2017*).

Corresponding authors
Lei Xia, pathology001@sina.com
Hairong Zhang,
sdzhhr7211@163.com

As a dynamically reversible modification process, m6A can be added by methyltransferases (also known as writer) and removed by demethylases (also known as eraser) (*Ma & Ji, 2020*). In addition, specific m6A recognition proteins (also known as readers) can directly or indirectly bind m6A sequences to affect RNA function (*Sun, Wu & Ming, 2019*; *Zhang et al., 2022b*). Unlike previously known modifications that are inherently irreversible, m6A modifications are reversible, giving this mode of modification the additional flexibility needed to regulate gene expression (*Ma & Ji, 2020*). Numerous studies have shown that m6A modifications are likely to be key regulators of various post-transcriptional gene regulatory processes (*Zhao, Roundtree & He, 2017*), which is of great significance in malignant tumors (*Wang et al., 2020b*; *He et al., 2019*), metabolic disease (*Kumari et al., 2021*), psychiatric disorders (*Han et al., 2020a*), and cardiovascular disease (*Qin et al., 2020*).

KIAA1429 (vir-like m6A methyltransferase associate, also known by VIRMA) is an isoform of methyltransferases, as the largest known protein in the methyltransferase complex, it can recruit catalytic core components (*e.g.*, METTL3/ METTL14/ WTAP) to guide regioselective m6A methylation (*Zhao & Xie, 2021*). In addition, m6A levels decrease most obviously in KIAA1429 knockdown cells, but not in METTL3 or METTL14 knockdown cells, indicating that KIAA1429 has a significant role in m6A modification (*Yue et al., 2018*; *Han et al., 2022*). KIAA1429 has important implications in malignant tumors such as breast cancer (*Qian et al., 2019*; *Liu et al., 2019*), hepatocellular carcinoma (HCC) (*Lan et al., 2019*; *Wang et al., 2020a*; *Liu et al., 2021*), non-small cell lung cancer (NSCLC) (*Xu et al., 2021*; *Tang et al., 2021*), colorectal cancer (CRC) (*Zhou et al., 2022*; *Ma et al., 2022*), osteosarcoma (OS) (*Han et al., 2020b*) , gastric cancer (GC) (*Yang et al., 2021*; *Miao et al., 2020*), and prostate cancer (PCA) (*Barros-Silva & Lobo, 2020*) . However, there are great differences in their effects and mechanisms, which can affect tumor development by whether depending on m6A modification or not (*Jiang et al., 2021*; *Huang, Weng & Chen, 2020*). Meanwhile, KIAA1429 participates in the occurrence and development of non-neoplastic diseases such as reproductive system disease (*Hu et al., 2020*) , cardiovascular system disease (*Wang et al., 2021*), respiratory system disease (*Fei et al., 2022*; *Dai et al., 2021*), and orthopedic diseases (*Zhu et al., 2021*) (Fig. 1). In this article, we reviewed the recent research progress on KIAA1429 dysregulation and its biological role in various human diseases and its underlying mechanisms.

## SURVEY METHODOLOGY

In this review, we comprehensively studied KIAA1429 dysregulation and its biological role in various human diseases. Experimental articles related to KIAA1429 are integrated in this article, and bioinformatic articles are excluded. Current research on KIAA1429 is mainly focused on tumorigenesis, and other diseases, especially those associated with developmental abnormalities, may also be associated with KIAA1429 abnormalities, which is also an area we should focus on.

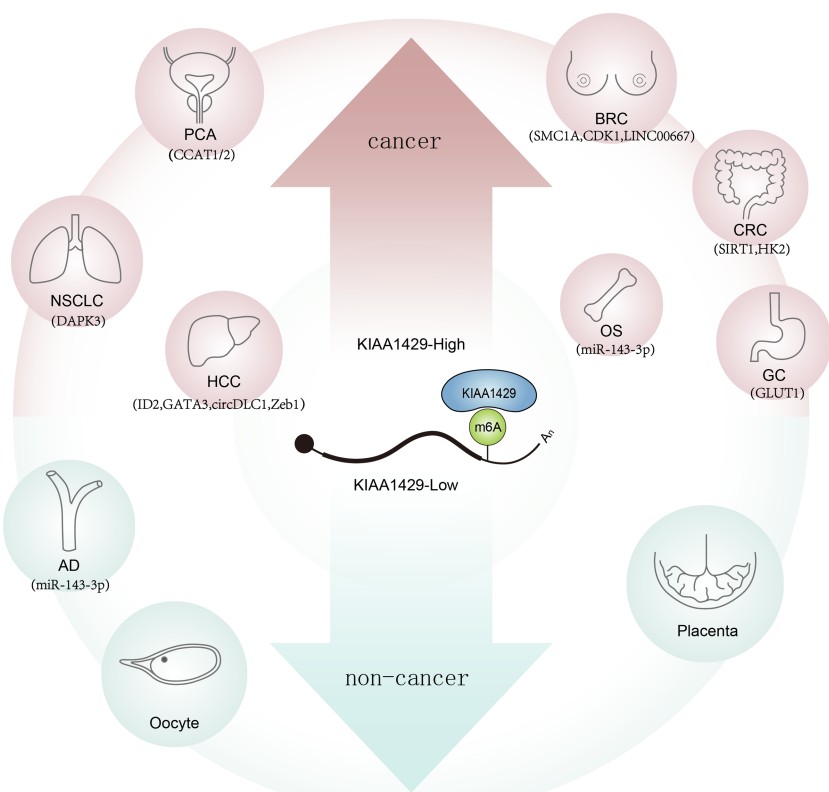

**Figure 1** **Role and potential targets of KIAA1429 in human diseases.** With the increase of KIAA1429 expression (red arrow), the tumorigenesis of the related tumor was promoted, while the decrease of KIAA1429 expression (blue arrow) induced the related diseases.

## KIAA1429 dysregulation and cancer
### KIAA1429 and breast cancer

m6A RNA methylation regulators have prognostic significance in breast cancer (*Zhang, Gu & Jiang, 2020*). *Zhang et al. (2022a)* found that KIAA1429 could significantly promote the migration and invasion of breast cancer cells. *In vitro* and *in vivo* experiments have shown that knockout of KIAA1429 gene inhibited tumor invasion and metastasis, and the mechanism might be related to downregulation of SNAIL (snail family transcriptional repressor 1) expression and EMT (epithelial-to-mesenchymal transition). Up-regulation of SNAIL expression prevented the inhibition of tumor cell migration, invasion and EMT progression caused by KIAA1429 knockout. However, KIAA1429 does not directly affect the expression of SNAIL, but indirectly affects the transcription and translation of SNAIL through SMC1A (Structural Maintenance Of 3 Chromosomes 1A).

*Qian et al. (2019)* reported that KIAA1429 was highly expressed in breast cancer tissues compared with normal breast tissues, and the overall survival of breast cancer patients with high KIAA1429 expression was significantly shorter than those with low KIAA1429 expression. KIAA1429 has been implicated in the proliferation and metastasis of breast cancer *in vivo* and *in vitro*. Cyclin-dependent kinases1 (CDK1) was identified as a potential

targeted gene of KIAA1429 in breast cancer by methylated RNA immunoprecipitation sequencing (MeRIP-seq) technology. In addition, 5-fluorouracil was found to be very effective in reducing the expression of KIAA1429 and CDK1 in breast cancer.

*Saiyu et al., (2022)* found that LINC00667 was a downstream target of KIAA1429 modification, and LINC00667 was upregulated after KIAA1429 overexpression. Mechanism analysis showed that KIAA1429 targeted the m6A modification site of LINC00667 and enhanced the stability of its mRNA. LINC00667 promotes the proliferation and migration of BC cells, and the high expression of LINC00667 is negatively correlated with the prognosis of BC patients.

### KIAA1429 and NSCLC

KIAA1429 has been found to be highly expressed in NSCLC patients and negatively correlated with prognosis. *In vitro* and *in vivo* experiments have suggested that high KIAA1429 expression could promote cell proliferation and tumor growth, whose mechanism was related to the m6A modification of death-associated protein kinase 3 (DAPK3), DAPK3 is a tumor suppressor gene of NSCLC, the cancer-promoting effect of KIAA1429 can be reversed when DAPK3 is highly expressed. And the process depended on YTHDF2/3 (*Xu et al., 2021*), The overexpression of YTHDF2/3 can abolish the proliferation inhibition and invasion reduction that is induced by the deficiency of KIAA1429. Thus, KIAA1429 relied on YTHDF2/3 to regulate cell proliferation, migration and invasion in H520 and A549 cells.

m6A modification plays an important role in drug resistance development in NSCLC. *Tang et al. (2021)* screened and identified the high expression of KIAA1429 in gemfibro-resistant NSCLC, indicating that knockout of KIAA1429 could not only inhibit the growth of PC9-GR cells *in vivo*, but reduced the IC50 value of PC9-GR cells. *Tang et al. (2021)* analyzed H1299, A549, PC9, and PC9-GR cells, and found that PC9-GR cells were KIAA1429 highly expressing cell lines and chemotherapy-resistant cell lines. Therefore, this cell line was selected for subsequent studies, then they found that KIAA1429 expression was inhibited. It can effectively reverse the gefifitinib of PC9-GR Drug resistance.

### KIAA1429 and HCC

KIAA1429 is closely associated with HCC, but the focus on the downstream targets of KIAA1429 varies among studies.

*Cheng et al. (2019)* conducted bioinformatic analysis by The Cancer Genome Atlas (TCGA) database and sealed differences in the expression of KIAA1429 between HCC and normal liver tissues. *In vitro* assays, the authors found that KIAA1429 promotes HCC cell proliferation, migration, and invasion. Then the author analyzed the genes that are differentially expressed and contain alterative m6A deposition after the interference with KIAA1429 expression in Gene Expression Omnibus database (GSE102493). And they found a significant negative correlation between KIAA1429 and DNA-binding protein inhibitor (ID2) in subsequent KEGG functional enrichment analysis, Finally, To further delineate the action mechanism between KIAA1429 and ID2, they built a MeRIP-PCR assay and was discovered that KIAA1429 facilitated migration and invasion of HCC by inhibiting ID2 *via* upregulating m6A modification of ID2 mRNA. *Kim et al. (2021)* found

that the proliferation, invasion and metastasis ability of the cells were significantly increased after ID2 expression was inhibited. However, high expression of KIAA1429 can inhibit ID2 expression, which may be one of the mechanisms of KIAA1429 promoting HCC development.

*In vitro* and *in vivo* experiment (*Lan et al., 2019*) displayed that knockout of KIAA1429 inhibited cell proliferation and metastasis, and its mechanism might be associated with GATA binding protein 3 (GATA3) pre-mRNA.

*Liu et al. (2021)* documented that circDLC1 (*i.e.*, exons 14, 15, and 16 from the DLC1 gene) was an important downstream target of KIAA1429 regulation. *In vitro* and *in vivo* experiments showed that overexpressed circDLC1 prevented the proliferation and metastasis of HCC and was positively correlated with the prognosis of HCC patients. Further studies demonstrated that circDLC1 interacted with RNA-binding protein HuR and blocked the interaction between HuR and MMP1 mRNA, thus circDLC1-HuR-MMP1 might be a potential therapeutic target.

*Wang et al. (2020a)* found the up-regulation of hsa_circ_0084922 from KIAA1429 in HCC cells and tumor tissues, and named it circKIAA1429. Overexpression of circKIAA1429 could promote the migration, invasion and EMT process of HCC. Zinc finger E-box-binding homeobox 1 (Zeb1) was also found to be a downstream target of circKIAA1429, and its up-regulation was involved in circKIAA1429-induced metastasis of HCC cells. The process required the participation of YTHDF3.

### KIAA1429 and colorectal cancer

In recent years, the clinical significance of M6A in colorectal cancer has attracted people's attention (*Ji et al., 2020b*). *Zhou et al. (2022)* analyzed data from TCGA and GEPIA database, finding the high expression of KIAA1429 in colorectal cancer. Subsequently, immunohistochemistry, Western Blot, and QRT-PCR were used for validation in order to confirm KIAA1429 overexpression in colorectal cancer specimens and cell lines. Further mechanistic analysis suggested that KIAA1429 regulated the mRNA stability of silent information regulator 2 homolog 1 (SIRT1) in a m6A-dependent manner, thereby elevating SIRT1 expression. Silencing information regulator 1 (SIRT1), a member of the HDAC family, is highly evolutionarily conserved histone and nonhistone deacetylase. The expression of SIRT1 is positively correlated with tumor growth, chemo-resistance, and metastasis (*Chen et al., 2012*; *Zhang et al., 2019*). *In vivo* experiments also showed that inhibition of KIAA1429 significantly inhibited the growth of colorectal tumors.

*Li et al. (2022)* studied the role of KIAA1429 in CRC. The results showed that KIAA1429 upregulation was closely related to poor prognosis of CRC patients. Biological function analysis showed that KIAA1429 promoted aerobic glycolysis, including glucose uptake, lactateproduction, ATP production and extracellular acidification rate (ECAR). Mechanistically, KIAA1429 increases the stability of HK2 (hexokinase 2) mRNA by binding to the m6A site of HK2 mRNA, thus positively upregulating HK2 level.

### KIAA1429 and osteosarcoma

m6A is a pivotal epitranscriptomic modification in common orthopaedic diseases (*Li et al., 2020*). Several studies have revealed the underlying molecular mechanisms of m6A

modifications in cancer (*Han et al., 2019*; *Liu et al., 2020*). *Han et al. (2020b)* revealed a significant overexpression of KIAA1429 in osteosarcoma which was strongly linked to poor prognosis. KIAA1429 silencing attenuated the proliferation, migration and invasion ability of osteosarcoma *in vitro*, as well as tumor growth *in vivo*. Mechanistically, miR-143-3p was considered to be a key specific mediator of KIAA1429 expression in osteosarcoma cells. In addition, knockout of KIAA1429 or overexpression of miR-143-3p could inhibit cancer stem cell properties. Mir-143-3p has been reported to be a suppressive miRNA in a variety of cancers, such as lung cancer, breast cancer, cervical cancer, and prostate cancers (*Gao et al., 2010*; *Liu et al., 2012*; *Xu et al., 2011*; *Osaki et al., 2011*; *Yan et al., 2014*). In addition, miR-143-3p plays a tumor suppressor role in osteosarcoma by targeting Bcl-2 and FOSL2 (*Sun et al., 2018*; *Li et al., 2016*). However, the molecular mechanism by which the miR-143-3p-KIAA1429 axis promotes cell proliferation and invasion remains to be further elucidated.

### KIAA1429 and gastric cancer

m6A-related genes were dysregulated in GC and were closely associated with prognosis of GC patients (*Guan et al., 2020*). *Yang et al. (2021)* indicated an elevated expression of KIAA1429 in gastric cancer, and its overexpression by tumor cells significantly up-regulated Lnc RNA LINC00958 expression. In addition KIAA1429 was positively correlated with LINC00958 expression in gastric adenocarcinoma, and high LINC00958 expression could reduce survival of patients with gastric cancer. The mechanism might be related to the stability of Glucose transporter-1 (GLUT1) mRNA, which in turn positively regulated aerobic glycolysis in gastric cancer.

### KIAA1429 and prostate cancer

Recently, it has been found that most of m6A methylation regulators were highly expressed in aggressive prostate cancer (*Ji et al., 2020a*). *Barros-Silva & Lobo (2020)* found that compared with normal cells or tissue, m6A RNA methylation levels in androgen-independent prostate cancer were significantly higher, accompanied by KIAA14291429 overexpression. Notably, KIAA1429 had a higher amplification and expression rate compared to other core subunits, probably because the KIAA1429 genome localized to chromosome 8q, who was commonly mutated in advanced and metastatic prostate cancer. KIAA1429 knockdown significantly inhibited the viability and proliferative capacity of PC-3 cells, reducing the malignant phenotype by weakening migratory and invasive properties. The mechanism might be related to m6A modification of lncRNACCAT1/2, which in turn promoted MYC expression.

## KIAA1429 and non-neoplastic diseases

The involvement of m6A in the methylation of RNA regulation during oocyte maturation has been proved (*Qi et al., 2016*). *Hu et al. (2020)* have demonstrated that KIAA1429-specific defects in oocytes can lead to follicular development defects and female infertility. Deficiency of KIAA1429 alters the expression pattern of oocyte-derived factors that coordinate follicle development and leads to GVBD defects. Oocyte growth was accompanied by accumulation and post-transcriptional regulation of a large number of

RNAs, and loss of KIAA1429 also contributed to abnormal RNA metabolism in GV oocytes. RNA sequence analysis revealed that loss of KIAA1429 altered the expression pattern of oocyte-derived factors necessary for follicular development. In addition, experiments have reported that loss of KIAA1429 reduced the level of m6A in oocytes, mainly affecting the alternative splicing of genes involved in oocytes development. Therefore, *Hu et al. (2020)* suggested that m6A methyltransferase KIAA1429 mediated RNA metabolism was playing a crucial role in folliculogenesis and oocyte competence maintenance. Furthermore, *Shen et al. (2022)* clearly showed that m6A was significantly suppressed due to the down-regulated expression of m6A transcription factors such as KIAA1429 in obese placentas. The authors found that the expression of WTAP, RBM15B and KIAA1429 genes significantly down-regulated m6A in placenta of obese pregnant women, and the mechanism was related to placental hypoxia. Due to the decreased level of m6A modification in placental tissue, the function of various mRNA may be impaired, but the specific mechanism has not been further studied.

*Wang et al. (2020a)* found a down-regulated KIAA1429 expression but up-regulated ALKBH5 expression in aortic tissues of patients with aortic dissection (AD). In addition, KIAA1429 and ALKBH5 could counter regulate the proliferation, HAEC apoptosis, and AD progression of HASMC cells from vascular-injected angiotensin II mice. *Wang et al. (2020a)* also found that 13 miRNAs were positively correlated with KIAA1429 while 12 miRNAs were negatively correlated with ALKBH5, in which knockout of overexpressed KIAA1429 or ALKBH5 could only significantly increase miR-143-3p. miR-143-3p could act by binding through the 3′ UTR region of DDX6. Moreover, KIAA1429/ALKBH5 could also produce marked effects by directly acting on this region.

## Mechanism of KIAA1429

KIAA1429 can affect related RNAs in m6A dependent and m6A independent ways, such as mediating alternative splicing, RNA maturation, RNA translation, RNA degradation, RNA stability, etc., and it can not only affect the function of mRNA (*Tang et al., 2021*; *Yang et al., 2021*; *Cheng et al., 2019*), but also participate in the occurrence and development of various diseases by affecting the function of lncRNA (*Osaki et al., 2011*) and circRNA (*Wang et al., 2020a*; *Liu et al., 2021*) (Fig. 2).

### KIAA1429 and long non-coding RNA

LncRNAs themselves had m6A methylation modifications site, and the number of modification sites on lncRNAs was often more than those m6A on mRNAs (1–5 sites), suggesting the important role of m6A methylation in both lncRNAs and its downstream regulation function (*Ransohoff, Wei & Khavari, 2018*).

In gastric cancer (*Yang et al., 2021*), MeRIP-Seq assays showed that KIAA1429 played its role through catalyzing the m6A modification site on lncRNA LINC00958. *In vitro* experiments indicate that LINC00958 promoted aerobic glycolysis in gastric cancer cells. m6A-modified LINC00958 could interact with GLUT1 mRNA and enhance GLUT1 mRNA transcriptional stability, thereby positively regulating aerobic glycolysis in gastric cancer.

In prostate cancer (*Barros-Silva & Lobo, 2020*), the cancer-promoting effect of KIAA1429 was associated with lncRNAs CCAT1 and CCAT2 m6A methylation modifications. CCAT,

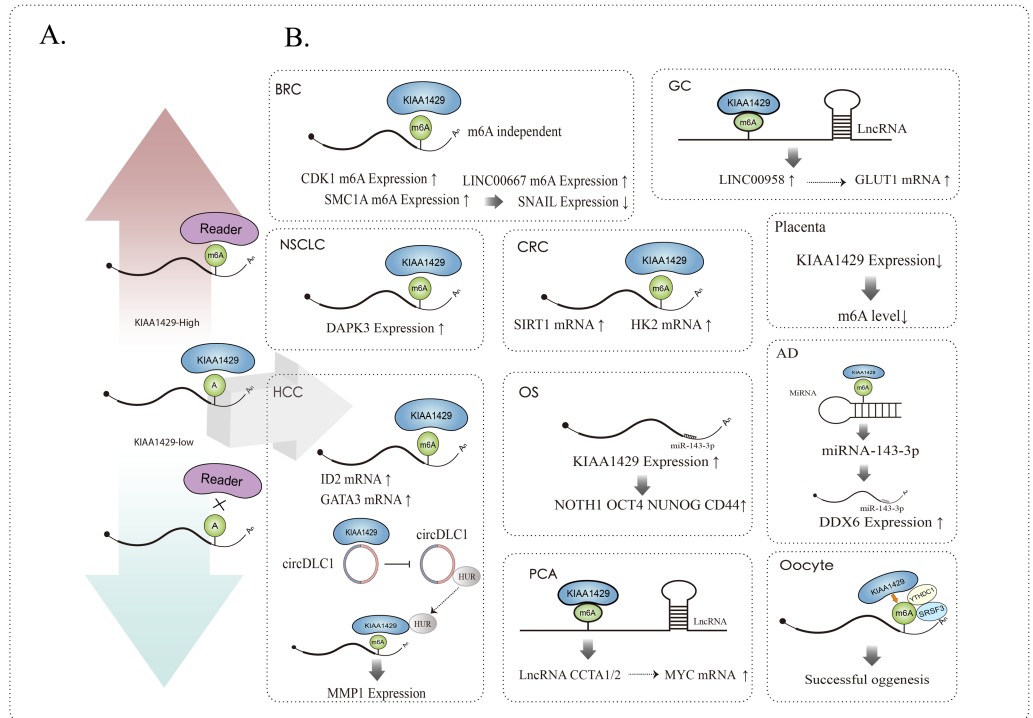

**Figure 2** (A) KIAA1429 expression and m6A modification, up regulation (red arrow) or down regulation (blue arrow). (B) KIAA1429 affects the expression and function of mRNA, circRNA, lncRNA, miRNA, *etc.* through m6A dependent or independent methods.

as a microRNA sponge, inhibits both let-7A and miR-145 by binding them, thereby inhibiting MYC. Among them, the increased methylation level of CCAT2 maintained its stability, thereby avoiding cleavage, while there was a significant positive correlation between lncRNA CCAT1/2 and MYC transcription. The lncRNA CCAT1/2 amplified the expression level of MYC in cancer cells by two different mechanisms which were directly and indirectly. On the one hand, it acted as a super-enhancer to positively regulate MYC mRNA, while on the other hand, it indirectly affected MYC through microRNAs let-7A and miR-145.

As mentioned above (*Saiyu et al., 2022*), LINC00667 is a downstream target of KIAA1429 in breast cancer. Overexpression of KIAA1429 can increase the m6A level of LINC00667 and enhance its stability, thereby increasing LINC00667 expression. LINC00667 can promote the proliferation and migration of BC cells, and the high expression of LINC00667 is negatively correlated with the prognosis of BC patients.

### KIAA1429 and miRNA

It is well-known that certain miRNAs have m6A modification sites, while miRNAs can also intervene in the expression of m6A modification-related proteins. Previous study has noted that the process of overexpression of KIAA1429 in osteosarcoma tissues was regulated by miR-143-3p. The miR-143-3p/KIAA1429 axis maintains the characteristics of

osteosarcoma stem cells through constitutive activation of notch signaling (*i.e.*, increased Notch1, Oct4, Nanog, CD44 protein expression) (*Han et al., 2020b*).

### KIAA1429 and mRNA

*KIAA1429 affects the translation of mRNA through m6A modification.* KIAA1429-mediated m6A modification occurred mainly in the 3′-UTR region and the termination codon region. *Xu et al. (2021)* focused on NSCLC and found that methylation modification of the 3′-UTR region by KIAA1429 was associated with YTHDF2/3. While Yue et al.'s study of Hela cells showed that KIAA1429 had influence on m6A modification of mRNA through recruiting the methyltransferase core component METTL3/METTL14/WTAP and interacting with polyadenylate cleavage factors CPSF5 and CPSF6 (*Yue et al., 2018*).

DAPK3: *Xu et al. (2021)* used KIAA1429 knockout and MeRIP-seq technology, identifying the close relationship between m6A modification of six genes in and KIAA1429, among which DAPK3 was the only up-regulated gene in the three NSCLC cell lines under its study. DAPK3 was involved in the regulation of apoptosis, autophagy, transcription, translation and actin cytoskeleton reorganization as a serine/threonine kinase. DAPK3 was also involved in the regulation of smooth muscle contraction, including type I (caspase-dependent) apoptotic signaling and type II (caspase-independent) autophagic cell death signaling, depending on the cellular environment.

ID2: KIAA1429 promoted migration and invasion of HCC by inhibiting ID2 expression through up-regulation of m6A-modified ID2 mRNA. ID2 acted as a transcriptional regulator (lacking the basic DNA binding domain) and negatively regulated basic helix-loop-helix (bHLH) transcription factors by forming heterodimers to inhibit DNA binding and transcriptional activity of bHLH transcription factors. ID2 was also involved in the regulation of a variety of cellular processes regarding cell growth, senescence, differentiation, apoptosis, angiogenesis, and tumor transformation (*Cheng et al., 2019*).

Homeobox protein (HOX1): HOX proteins were major regulators of embryonic development and produce a marked effect in tumorigenesis (*Belpaire et al., 2021*). The development of resistance of KIAA1429 to gemfibrozil might be related to HOX1 among NSCLC patients. HOX1 was a sequence-specific transcription factor belonging to developmental regulatory system. KIAA1429 enhanced the stability of its mRNA by m6A modification of the 3′-UTR of HOXA1. Knockout of HOXA1 could similarly inhibit the transfer of PC9-GR cells and resistance to metastasis. KIAA1429/HOXA1 therefore played an important role in tumor formation and drug resistance development (*Tang et al., 2021*).

GATA3: GATA3 was identified as a direct downstream target of KIAA1429 mediated m6A modification (*Lan et al., 2019*). KIAA1429 induced m6A methylation on the 3′-UTR of GATA3 mRNA, resulting in the separation and degradation of GATA3 mRNA from the RNA-binding protein HuR. In addition, KIAA1429 induced m6A methylation on the 3′-UTR of GATA3 pre-mRNA, resulting in the separation of the RNA-binding protein HuR from GATA3 pre-mRNA and the degradation of GATA3 mRNA. At the same time, previous results confirmed that LncRNA GATA3-AS who was transcribed from the antisense strand of the GATA3 gene, acted as a cis-acting element in the preferential interaction between

KIAA14291429 and GATA3 pre-mRNA. GATA3-AS knockdown significantly inhibited the malignant phenotype of HCC cells, while inhibition of GATA3 could rescue the malignant phenotype of HCC cells.

SIRT1: SIRT1 was a protein deacetylase containing a highly conserved sequence and was classified as histone deacetylase (HDACs) III (*Chen et al., 2012*; *Zhang et al., 2019*). In colorectal cancer, SIRT1 has been reported to be involved in tumor development and its expression was positively correlated with tumor progression in clinical practice (*Shen et al., 2016*; *Chen et al., 2014*; *Lv et al., 2014*). *Zhou et al. (2022)* believe that KIAA1429 promotes colorectal cancer and is related to SIRT. Through MeRIP-seq experiments, it was found that only SIRT1 transcripts could be immunoprecipitated by KIAA1429. KIAA1429 regulated the mRNA stability of SIRT1 in a m6A-dependent manner, thereby elevating SIRT1 expression.

*KIAA1429 affects the translation of mRNA in a manner independent of m6A modification.* SMC1A: *Zhang et al. (2022a)* found that KIAA1429 could enhance the stability of SMC1A mRNA by directly targeting the motif of SMC1A mRNA, which in turn increased the expression of SMC1A protein. SMC1A was involved in chromosome cohesion during cell cycle and DNA repair and was a central component of the cohesive complex. However, instead of altering the degree of m6A of SMC1A mRNA, KIAA1429 directly bounded to a specific motif in the 3′-UTR of SMC1A mRNA, thereby enhancing the stability of SMC1A mRNA. SMC1A promoted SNAIL expression by directly binding to the promoter region of the SNAIL gene. SNAIL was involved in the induction of EMT, embryonic mesoderm formation and maintenance, growth arrest, survival, and cell migration, all of which could promote the migration and invasion of breast cancer cells.

CDK1: CDKs were a family of proteins involved in cell cycle regulation that were frequently highly expressed or mutated in various cancers (*Huang et al., 2021*; *Izadi et al., 2020*). Immunohistochemistry analysis revealed a significant association of KIAA1429 with CDK1 in breast cancer tissue microarrays and *in vivo* xenograft animal models. KIAA1429 delayed the half-life of CDK1 mRNA by increasing its stability in a m6A-independent manner of m6A through its interaction with CDK1 mRNA (*Qian et al., 2019*).

HK2: *Zhang et al. (2019)* suggested that KIAA1429 upregulation is closely related to poor prognosis of CRC patients. Mechanistically, KIAA1429 can bind to the 3-UTR site of HK2 mRNA and increase the mRNA stability of HK2 through m6A independent, thereby upregulating the expression level of HK2.

### KIAA1429 and circRNA

CircDLC1 expression was decreased in HCC tissues, and overexpression of circDLC1 inhibited the proliferation and viability of HCC cells *in vitro* and *in vivo*, while silencing circDLC1 played the opposite role. *Liu et al. (2021)* confirmed that circDLC1 (exons 14, 15, and 16 from the DLC1 gene) was a downstream target of KIAA1429 regulation, which was positively correlated with prognosis. CircDLC1 interacted with the RNA-binding protein HuR while blocking the interaction between HuR and MMP1 mRNA, thereby reducing MMP1 expression.

CircKIAA1429 was able to accelerate the progression of HCC, and Zeb1 was a downstream target of circKIAA1429. As a transcriptional repressor, first Zeb1 could inhibit IL-2 gene expression and enhance or inhibit the promoter activity of the ATP1A1 gene depending on the number of cDNAs and cell type. Second, Zeb1 acted on the E-cadherin promoter and induced EMT by recruiting SMARCA4/BRG1. Third, Zeb1 inhibited transcription of BCL6 in response to the corepressor CTBP1. YTHDF3 stabilized zeb1 through m6A modification (*Wang et al., 2020a*).

## DISCUSSION

Recently, the role of m6A modification abnormalities in diseases has gradually attracted people's attention. The m6A modification refered to methylation of the N6 position of adenosine base (*Zhang et al., 2021*), which can regulate the structure, stability, splicing, export, transcription and decay of mRNA, miRNA and lncRNA through methyltransferases, demethylases and m6A binding proteins, and then widely affect the life processes of various organisms (*Tian et al., 2020*; *Song et al., 2021*). KIAA1429 is a major m6A methyltransferase and the largest protein of the methyltransferase complex (*Yue et al., 2018*). The attention was first focused on KIAA1429 because of the discovery of its ortholog, which was shown to interact with Drosophila WTAP in the context of sex-specific splicing (*Ortega et al., 2003*).

Although some studies have suggested that elevated KIAA1429 expression could inhibit the occurrence of thyroid cancer (*Ransohoff, Wei & Khavari, 2018*), its results were mainly based on bioinformatic analysis which was not included in this article. This review integrates the experimental articles related to KIAA1429, and excludes the articles that contain bioinformatics analysis but lack experimental verification. Previous studies have shown that increased expression of KIAA1429 may induce tumorigenesis. While the role acts oppositely in non-neoplastic diseases, where reduced KIAA1429 expression can lead to germ cell dysplasia and AD (Table 1).

For different cancers with different genetic backgrounds, m6A RNA methylation can control cancer progression by regulating oncogene expression, cancer cell differentiation, proliferation, migration, angiogenesis and tumor microenvironment. Therefore, targeting m6A RNA modifiers can provide potential therapeutic targets for various human cancers. For example, recent studies have shown that R-2-hydroxyglutarate (R-2HG) can inhibit leukemic cell proliferation and induce apoptosis by targeting FTO/m6A/MYC/CEPA signaling (*Su et al., 2018*). A non-steroidal anti-inflammatory drug named meclofenamic acid (MA) has recently been identified as a selective inhibitor of FTO (*Huang et al., 2015*). FB23 and FB23-2 are two promising FTO inhibitors, which can selectively inhibit the m6A demethylase activity of FTO, thereby significantly inhibiting the proliferation and promoting apoptosis of AML cells (*Huang et al., 2019*). In addition, two compounds known as CS1 and CS2 were shown to bind tightly to the FTO protein and block its catalysis, thus exhibiting potent antitumor effects in several types of cancer (*Su et al., 2020*). Mo-i-500 was recently found to be a selective inhibitor of FTO, which inhibited the proliferation of triple-negative breast cancer cells (*Singh et al., 2016*). However, the drug research targeting KIAA1429 has not been reported yet. This is also one of our future research directions.

**Table 1  Expression, study design, and biological function of KIAA1429 in various diseases.**

| Disease | Expression | Study Design | Role | Biological Function | Target | References |
| --- | --- | --- | --- | --- | --- | --- |
| BRC | Upregulated | Cell lines, Animal models | Oncogene | Migration, Invasion, EMT | SMC1A | *Zhu et al. (2021)* |
| BRC | Upregulated | Cell lines, Human samples, Animal models | Oncogene | Proliferation, Metastasis | CDK1 | *Han et al. (2022)* |
| BRC | Upregulated | Human samples, cell lines | Oncogene | Proliferation, Migration | LINC00667 | *Saiyu Ren, Liu & Zhang (2022)* |
| NSCLC | Upregulated | Human samples, cell lines | Oncogene | Proliferation | DAPK3 | *Liu et al. (2021)* |
| HCC | Upregulated | Human samples, Cell lines | Oncogene | Migration, Invasion | ID2 | *Zhang, Gu & Jiang (2020)* |
| HCC | Upregulated | Human samples, Cell lines | Oncogene | Proliferation, Metastasis | GATA3 | *Liu et al. (2019)* |
| HCC | Upregulated | Cell lines, Human samples, Animal models | Oncogene | Proliferation, Metastasis | circDLC1 | *Wang et al. (2020a)* |
| HCC | Upregulated | Cell lines, Human samples, Animal models | Oncogene | Migration, Invasion, EMT | Zeb1 | *Lan et al. (2019)* |
| CRC | Upregulated | Cell lines, Animal models | Oncogene | Proliferation | SIRT1 | *Tang et al. (2021)* |
| CRC | Upregulated | Cell lines, Human samples, Animal models | Oncogene | Aerobic Glycolysis | HK2 | *Ji et al. (2020b)* |
| OS | Upregulated | Cell lines, Human samples, Animal models | Oncogene | Proliferation, Migration, Invasion | miR-143-3p | *Ma et al. (2022)* |
| GC | Upregulated | Cell lines, Human samples, Animal models | Oncogene | Metabolic reprogramming | GLUT1 | *Han et al. (2020b)* |
| PCA | Upregulated | Human samples | Oncogene | Proliferation, Migration, Invasion | CCAT1/2 | *Miao et al. (2020)* |
| Oocyte | Downregulated | Cell lines, Animal models | Suppressor | Folliculogenesis, Maintenance of oocyte competence | – | *Huang, Weng & Chen (2020)* |
| AD | Downregulated | Cell lines, Human samples, Animal models | Suppressor | HASMC proliferation, HAEC apoptosis, and AD progression | miR-143-3p | *Wang et al. (2020a)* |

Therefore, we believe that KIAA1429 overexpression should be a predictor of poor prognosis in tumors, and low expression is a predictor of poor prognosis in non tumor diseases, and similar conclusions were obtained by bioinformatics analysis, but the related clinical data need to be further collected and analyzed. Current research on KIAA1429 is mainly focused on tumorigenesis, and other diseases, especially those associated with developmental abnormalities, may also be associated with KIAA1429 abnormalities, which is also an area we should focus on.

## ACKNOWLEDGEMENTS

Over the course of my researching and writing this article, I would like to express my thanks to all those who have helped me.

### Funding

This work was supported by the National Natural Science Foundation of China (No. 82104554). The funders had no role in study design, data collection and analysis, decision to publish, or preparation of the manuscript.

### Grant Disclosures

The following grant information was disclosed by the authors:
National Natural Science Foundation of China: 82104554.

### Competing Interests

The authors declare there are no competing interests.

### Author Contributions

- Xiaoyu Zhang conceived and designed the experiments, performed the experiments, prepared figures and/or tables, and approved the final draft.
- Meng jiao Li analyzed the data, authored or reviewed drafts of the article, and approved the final draft.
- Lei Xia analyzed the data, authored or reviewed drafts of the article, and approved the final draft.
- Hairong Zhang analyzed the data, authored or reviewed drafts of the article, and approved the final draft.

### Data Availability

There is no data and code; this article is a literature review.

### Supplemental Information

Supplemental information for this article can be found online at http://dx.doi.org/10.7717/peerj.14334#supplemental-information.

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
