# Peer review of "The biological function of m6A methyltransferase KIAA1429 and its role in human disease"

_PeerJ, doi:10.7717/peerj.14334_

## Round 0.1 · original submission · Major Revisions

Dear Dr. Zhang,

Thank you for submitting your manuscript "The biological function of m6A methyltransferase KIAA1429 and its role in human disease" to PeerJ. We have now received reports from the three reviewers and, after careful consideration, we have decided to invite a major revision of the manuscript.

As you will see from the reports copied below, the reviewers raise important concerns, especially with the grammatical errors and inconsistencies that are very hard to follow. We find that these concerns limit the strength of the study, and therefore we ask you to address them with additional work. Without substantial revisions, we will be unlikely to send the paper back for review.

If you feel that you are able to comprehensively address the reviewers’ concerns, please provide a point-by-point response to these comments along with your revision. Please show all changes in the manuscript text file with track changes or color highlighting. If you are unable to address specific reviewer requests or find any points invalid, please explain why in the point-by-point response.

My suggestion:
It’s better to have the manuscript reviewed by an English expert to improve its presentation.

Thanks

Abhishek Tyagi, PhD
Academic Editor,
PeerJ

Reviewer 1 ·

Basic reporting

The manuscript is poorly written and needs to be largely reworked. The organization of the manuscript is not clear and the arguments are hard to follow.

Major Problems:

1. Poor introduction and description of the manuscript.

2. Some of the seminal studies were not explained in the manuscript (Ex: References - 30, 33).

3. The figures were not informative. The figure legends were sketchy and lack description.

4. Exact sentences from lines 64-65 , and 18-19 were repeated in lines 67-68, and 68-69 respectively.

5. Grammatical and spelling mistakes were observed in several cases. Ex: In Figure 2, the authors mentioned “Nunog” instead of Nanog, as well as “oggenesis” instead of oogenesis.

Experimental design

No comments

Validity of the findings

No comments

Reviewer 2 ·

Basic reporting

The manuscript by Zhang X and Li M et.al. has summarized the recent understandings of methyltransferase KIAA1429, mostly in the context of cancer. The manuscript is overall well-written in professional English. The introduction is appropriate, and the review is well-referenced. The strength of the review is that it has summarized recent findings in the field. There are a couple of new studies that should be added to the review which has been mentioned in additional comments. There are a few typographical and grammatical errors that should be fixed by proof-reading.

Experimental design

Survey methodology has been explained properly. The reference literatures have been paraphrased properly, although some studies need to be further elaborated, by briefly stating the functions of the target molecules in the various disease contexts. The review has been organized logically, but the subsections should be more well-defined.

Validity of the findings

The outcomes have been properly explained and is linked to the studies that were referenced. The discussion section has to be significantly improved to indicate future directions. The authors should discuss more on the potential of using KIAA1429 as a biomarker for cancer. The authors should shed light on what are the methyltransferases currently used as successful biomarkers for cancer prognosis. The possibility of targeting KIAA1429 should also be touched upon.

Additional comments

I have a few major and minor concerns have been listed below:
• It is suggested that the references are mostly numbered according to the order that they appear in the review.
• Table1, Figure 1 and Figure 2 has not been cited in the manuscript. Figure 1 should be improved and made more informative. One suggestion for Figure1 would be include the known molecular targets and overall implications of high and low KIAA1429 in the different organs.
• A major concern is that the authors should indicate the organisms in which the observations were made. It is not clear from the review as to which studies were done in human cancer tissues and which studies were done in cell lines and animal models.
• Line 13: Also indicate that KIAA1429 is also known by VIRMA.
• In the role of KIAA1429 in breast cancer, the authors should add another study recently done which further supports the role of the methyltransferase in tumor progression. https://doi.org/10.1080/21655979.2022.2077893
• Please expand NSCLC, HCC once in the manuscript.
• Explain how YTHDF2/3 is important for the process in Line 94, and the function of YTHDF3 in line 127.
• Line 97-100: The authors are suggested to explain the observations more clearly. For example, what are the significance of PC9-GR cells?
• Line 109-113: Explain how does ID2 inhibit HCC, with references.
• Line 133: Explain briefly the general function and role of SIRT1 in colorectal cancer promotion.
• Please include another evidence for participation of KIAA1429 in colorectal cancer https://doi.org/10.1080/21655979.2022.2065952
• How does miR-143-3p influences osteosarcoma? The authors should explain with reference, atleast briefly before explaining the mechanisms in details later on.
• Line 173: What is the outcome of downregulated KIAA1429 in obese placenta? Please explain in details.
• Line 202: Does CCAT1/2 inhibits microRNAs let-7A and miR-145 ?
• Line 214-217: Please add reference for the study.
• The Acknowledgement is very elaborate and should be reduced in size, and should follow PeerJ format.

Reviewer 3 ·

Basic reporting

English language used in the review is poor, needs tremendous improvement. There are many grammatical mistakes throughout the review.

Experimental design

Literature has been thoroughly searched, however some minor weaknesses stem from the non-disease role of KIAA1429.

Validity of the findings

no comment

Additional comments

1) The most important issue is that there are so many grammatical mistakes and English language must be greatly improved to clearly understand the text for international audience. Some examples where the language can be improved include lines 72,19,20,72,162,300,301. Scientific language must be used to improve the quality of the review.
2) Authors mention dual effect of KIAA1429 in cancer in the abstract. However, in the whole review the authors talk about tumorigenic role of KIAA1429 in cancer. Authors need to clarify what dual effect means.
3) Authors have done good review of literature on the role of KIAA1429 role in tumorigenesis in multiple cancer types but there is not much focus on KIAA1429 role in non-neoplastic diseases which makes the review title misleading. Since KIAA1429 has been well studied in oncogenic transformation, it would be best for the authors to focus exclusively on the role of KIAA1429 in cancer.
4) The authors may include one section on the upstream signaling that regulate KIAA1429, which would help the audience and the field to find ways to target KIAA1429 in future research.
5) Discussion needs to be more elaborate, implications on the field and future directions. Since KIAA1429 has a potential role in promoting tumor growth, it should be best to discuss on ways the KIAA1429 can be targeted for clinical benefits other than just a prognostic factor
6) Abstract must be rewritten to emphasize the importance on the role of KIAA1429 in cancer should remove intended audience sentence.
7) Figure 1 does not add any extra knowledge to the review, which is same as that of Table 1, it can be removed

---

## Round 0.2 · accepted · Accept

Dear Dr. Zhang and Dr. Xia,

We are delighted to accept your manuscript entitled "The biological function of m6A methyltransferase KIAA1429 and its role in human disease" for publication in PeerJ. Thank you for choosing to publish your interesting work with us.


With kind regards,
Abhishek Tyagi
Academic Editor, PeerJ

Reviewer 1 ·

Basic reporting

No Comment

Experimental design

No Comment

Validity of the findings

No Comment

Reviewer 2 ·

Basic reporting

The concerns have been addressed.

Experimental design

The relevant concerns have been addressed.

Validity of the findings

The concerns have been addressed

Additional comments

All the concerns have been answered and incorporated appropriately in the manuscript.